# Immunoregulatory Functions of the IL-12 Family of Cytokines in Antiviral Systems

**DOI:** 10.3390/v11090772

**Published:** 2019-08-22

**Authors:** Yifei Guo, Wei Cao, Ying Zhu

**Affiliations:** State Key Laboratory of Virology, Modern Virology Research Center, College of Life Sciences, Wuhan University, Wuhan 430072, China

**Keywords:** interleukin (IL)-12 family, viral infection, immune systems

## Abstract

Members of the interleukin 12 (IL-12) family have been known to be inflammatory factors since their discovery. The IL-12 family consists of IL-12, IL-23, IL-27, IL-35, and a new member, IL-39, which has recently been identified and has not yet been studied extensively. Current literature has described the mechanisms of immunity of these cytokines and potential uses for therapy and medical cures. IL-12 was found first and is effective in combatting a wide range of naturally occurring viral infections through the upregulation of various cytokines to clear the infected cells. IL-23 has an essential function in immune networks, can induce IL-17 production, and can antagonize inhibition from IL-12 in the presence of T helper (Th) 17 cells, resulting in type II IFN (IFN-γ) regulation. IL-27 has a competitive relationship to IL-35 because they both include the same subunit, the Epstein–Barr virus-induced gene3 (EBi3). This review provides a simple introduction to the IL-12 family and focuses on their functions relevant to their actions to counteract viral infections.

## 1. Introduction

All members of the IL-12 cytokine family consist of two kinds of subunits: an α-and a β-cytokine subunit. Different combinations of these two kinds of subunits give rise to different cytokines, so that new members are constantly being discovered, such as IL-39.

Kobayashi et al. first discovered natural killer cell stimulatory factor (NKSF) in 1989 [1], which was initially named IL-12. IL-12 has a heterodimeric structure comprising an α-subunit (IL-12p35) and a β-subunit (IL-12p40). IL-12 recognizes its receptors, IL-12Rβ1 and IL-12Rβ2, leading to the binding of Non-receptor tyrosine-protein kinase (TYK2) and Janus kinase 2 (JAK2), respectively, mainly in order to induce signal transducer and activator of transcription 4 (STAT4), which is necessary for IL-12 function [2]. Many studies have clearly demonstrated that IL-12 enhances the connection between the innate and adaptive immune responses.

IL-23 was first described in 2000 by Oppman et al. Expression of IL-23 is strongly mediated through its two subunits (IL-23p19 and IL-12p40) because IL-23 shares IL-12p40 with IL-12. When IL-12p35 level is higher than that of IL-23p19, IL-23 production is suppressed [2]. Clusterdifferentiation 40 (CD40) and CD40 ligand (CD40L) stimulate the expression of the two subunits of IL-23 receptors (IL-12R-β1 and IL-23R), which bind to the same proteins (TYK2 and JAK2) as IL-12 to stimulate STAT3 and STAT4 [3].What is clear is that IL-23 resembles IL-12 not only in structure but also in function, because both of them are pro-inflammatory cytokines.

According to current studies, IL-27 is a pro-inflammatory cytokine as well as an anti-inflammatory cytokine within the IL-12 family, and antigen-presenting cells (APCs) are the main source of IL-27. IL-27p28 combines with a IL-27R (WSX-1) receptor and Epstein–Barr virus-induced gene3 (EBi3) matches with a gp130 receptor [3]. In this case, JAK1 cooperates with JAK2 to phosphorylate STAT1-STAT3 dimers [2].

IL-35 shares the IL-12p35 subunit with IL-12, and EBi3 with IL-27 [2]. Interestingly, IL-35 can interact with four different receptor combinations: gp130:IL-12β2, gp130:gp130, IL-12β2:IL-12β2, and IL-12β2:WSX-1 [4]. IL-35 causes STAT1 and STAT4 phosphorylation in T cells, while it activates STAT1 and STAT3 in B cells via JAK1 and JAK2 [2].

Due to the combination of different subunits, the IL-12 family also includes IL-39, which consists of IL-23p19 and EBi3, a pro-inflammatory cytokine [5]. Receptors for IL-39 are IL-23R and gp130, mainly activating STAT1 and STAT3, respectively [4]. Dendritic cells (DCs) and macrophages play significant roles in subunit expression, and Lipopolysaccharide-activated (LPS-activated) B lymphocytes are also useful for IL-39 production [5]. However, there are few studies on the anti-viral role of IL-39 in immune networks, and these studies only focus on animal models. Thus, the results cannot be reliably extrapolated to humans [6]. Since IL-39 was only discovered in 2019, its function in acting against infection is not discussed below.

As discussed above, all members of the IL-12 family are heterodimers and are only active when both subunits are present. IL-12 and IL-23 share one subunit, as do IL-27, IL-35, and IL-39, so these cytokines compete with each other for subunits and receptors. As a result, they may differ in production levels (Figure 1).

## 2. Anti-Viral Functions

### 2.1. IL-12

Various viruses are responsible for the induction of IL-12 expression, including hepatitis B virus (HBV), human immunodeficiency virus (HIV), human metapneumovirus (hMPV), and herpes simplex virus (HSV). When a virus enters a cell, it represents a pathogen-associated molecular pattern (PAMP) [7]. Then, Toll-like receptors (TLRs) send out signals to trigger antigen-presenting cells (APCs), including dendritic cells (DCs) and macrophages (MΦs), both of which can produce IL-12 [7]. With the exception of the aforementioned innate immunity, IL-12 also participates in the regulation of adaptive immunity. Naive CD4+T cells that have T-cell receptors (TCRs) and T-cell cytokine receptors (TCCRs) play roles in the differentiation of these cells with the help of T-bet and IL-27. Because Th1 cells bear various receptors, IL-12 and IL-23 can both promote the transformation of naïve Th1 cells into mature Th1 cells that induce IFN-γ, illustrating how type 1 immunity works [7]. Then, IL-12 induces IFN-γ secretion by STAT4 [8] and activating natural killer cell (NK cells) [9,10]. T-bet activity is enhanced in the presence of IFN-γ, resulting in the expression of IL-12Rβ2, thus permitting a Th1 response to IL-12 to develop. In this case, more IL-12 and IFN-γ can be utilized to augment the production of IL-12 by stimulating APCs, NK, and T cells [11]. Likewise, IL-12 ultimately accelerates IFN-γ production [12]. For example, during respiratory syncytial virus (RSV) infection in vitro, IL-12 mediates the transformation of Th17 cells into Th1 cells to induce IFN-γ production via an IL-23/IL-23R cascade [13]. Moreover, activating STAT4, the Jun oncogene (c-*Jun*), in cooperation with IFN promoters, can also boost IFN-γ expression and Th1 differentiation [2] (Figure 1). Furthermore, suppressor of cytokine signaling 3 (SOCS3) can potentially compete with JAK2 for STAT4 binding, and thus Th1 cells can also be inhibited [2].

In contrast to previous studies, recently published results show that both IL-12p35 and IL-12p40 are required in order to form IL-12p70. Without IL-12 p40, hMPV infection is more severe and the weight of infected mice is lower [14]. IL-12 p40 is necessary for regulating the cytokine response, and a deficiency of IL-12p40 leads to decreases in IFN-γ, which mediates Th1 cell activity [14]. Taking another example, recombinant HSV-IL-2 has a suppressive effect on IL-12p35 and IL-12p40, revealing a negative feedback mechanism including central nervous system (CNS) demyelination caused by autoimmune T cells. This kind of demyelination can be blocked by the transfer of macrophages infected with recombinant HSV-IL-12p40 or HSV-IL-12p70 virus but not when infected with HSV-IL-12p35 virus [15]. Additionally, the accumulation of d-2-hydroxyglutarate (d-2-HG) and l-2-hydroxyglutarate (l-2-HG) inhibits the secretion of both IL-12 subunits through monocyte-derived DCs (MDDCs) [16]. 

In HIV, tumor necrosis factor (TNF) family molecules work together to maximally enhance IL-12 expression [17], stimulating anti-HIV antibodies [18]. In HIV-1-infected MDDCs, members of the TNF superfamily are typically present, including CD40L, receptor activator of nuclear factor kappa-B (NF-κb) ligand (RANKL), and TNF-α. Among them, CD40L induces the highest increase in IL-12 expression to enhance inflammatory responses, but cooperation of these three factors results in higher expression [17]. Interestingly, CD40L also plays a significant role in promoting the recognition of programmed cell death ligand 1 (PD-L1) by programmed cell death 1 (PD-1), which ultimately leads to the progression of HIV-1 [19]. When PD-L1 and PD-1 react with each other, there is a negative effect on CD8 T cells, and the TCR signaling pathway is inhibited [20]. As a result, there is an increased expression of the two subunits of the IL-23 receptor so that IL-23 can compete with IL-12 for IL-12p40 to decrease IL-12 production levels. Obviously, anti-HIV-1 effects are strongly dampened. Therefore, to weaken HIV-1 infections, inhibiting PD-1/PD-L1 signals may be an effective strategy [19] (Table 1).

IL-12 has an apparent effect on suppressing HBV gene expression. Available research shows that T cell receptors, in the presence of substances that stimulate these receptors (such as CD28) and pro-inflammatory cytokines (such as IL-12), are indispensable for sending signals to trigger CD8 T cells that can specifically recognize HBV antigens. Similar to CD8 T cells, CD4 T cells are also activated when co-cultured with IL-12 and cause HBV clearance [21]. In addition, IL-12 inhibits PD-1 in this kind of infection, increases T-bet, and promotes IFN-γ and TNF-α production for immunity [22] (Table 1).

In conclusion, IL-12 induces many cytokines and cells modulating inflammatory responses and plays a vital role in type 1 immunity. 

### 2.2. IL-23

IL-23 induces type 3 immunity. IL-23 increases CD4+Th17 cell production, leading to IL-17 expansion with the help of STAT3 [20,23]. Interestingly, IL12 and IL-27 can both induce IL-10-producing type 1 regulatory cells (Tr1) that ultimately antagonize IL-23 [24]. In this condition, Th17 cells maintain the production of IL-23 to overcome negative feedbacks to IL-23 [8]. In turn, IL-23 antagonizes IL-12-promoting IFN-γ by blocking IL-12 signaling to phosphorylate STAT4, which can simultaneously lead to the inhibition of Th1 cell differentiation [25]. Furthermore, the stimulation of monocytes with LPS robustly promotes IL-23 secretion, causing NK cells to express more IL-23 receptors, and consequently IFN-γ is also downregulated [26]. A classic example can be seen in the process of HIV-1 infection, where IL-23 dramatically decreases IL-12-induced IFN-γ and simultaneously overcomes the inhibitory effect of regulatory T (Treg) cells. IL-23 can also vigorously induce phosphorylation of STAT3, while its induction of STAT4 is not as strong (Figure 1). As a result, naïve T cells differentiate into Th17 cells that lack the ability to produce IFN-γ [25]. Th17 cells can produce IL-17/22 and can clear infected cells as well as produce IL-23 to achieve Th17 polarization [27]. Since Th17 cells promote the production of Th1 cells, IL-23 markedly increases via myeloid dendritic cell (mDC) maturation with a reduction in IL-12p70 related to Th1 cell polarization after HIV infection [28]. Similarly, in order to inhibit IFN-γ production, IL-23 can promote SOCS1 production in T cells [28]. In vitro, the expansion of IL-23 can mediate the cooperation of T lymphocytes and alveolar macrophages (AM) to stimulate IL-23R expression in HIV-1+ smokers [29] (Table 1). In HCV infections, with the increase in IL-23, IL-17A as well as IFN-γ induced by peripheral blood mononuclear cells (PBMCs) increases while the number of PBMCs producing IL-21 decreases through the regulation of Th17 [30]. HIV and hepatitis C virus (HCV) are two kinds of RNA virus. A review by Zhu H et al. concludes that *DDX58*-induced retinoic acid-inducible gene I (RIG-I) is necessary for the recognition of RNA viruses that have specific ligands for RIG-I [31]. This gene product stimulates endogenous IL-23 production that is limited by nuclear factor kappa-B (NF-κb) instead of interferon regulatory factor 3/7 (IRF-3/7). IL-23 can also give a positive feedback to increase RIG-I production [31]. When HIV and HCV co-infect a cell, the induced IL-23 promotes IFN-α, and HCV can protect interferon alpha/beta receptor 2 (IFNAR2) receptors from being downregulated by IFN-α. In contrast, low concentrations of IL-23 can destroy Th17CD4+T cells [32]. 

As can be seen from the above description, IL-23 is available throughout a wide range of virus infections. Analysis of HBV-infected patients suggested the presence of IL-23 in serum or in hepatic tissue is closely associated with liver injury [33]. IL-23 mediates its downstream factor IL-17 in determining the indispensable role of γδT cells, affecting the extent of liver damage [34]. Increases in γδT cells are associated with decreased IFN-γ in CD4+T cells, improvement of cellular necrosis [34], attenuated STAT3 signaling, and overexpression of SOCS3 with T cell exhaustion due to PD-1 augmentation [35]. Simultaneously, a reduction in TNF-α can also be observed in HBV infection accompanied by liver injury, and consequently, IL-23-neutralizing antibodies are required for reducing liver damage [36]. The use of these antibodies in treatment is superior to the suppression of IL-17 in several ways, such as reducing the production of cytokines that are correlated with Th17 expression, controlling neutrophil chemoattractant in the liver, and suppressing stabilization factors [36] (Table 1). Without these antibodies, TLR8 stimulates human neutrophils to promote Th17 cell polarization by IL-23 [37]. In this process, IL-23 triggers IL-1R signaling and ultimately induces Th17 cells in cooperation with IL-27, IL-1β, and IL-6 [38]. Nevertheless, the immune system must mediate the balance between IL-35 and IL-12 because IL-23 can be pathogenic if it is overexpressed [25]. In addition, HBV infection is the main cause of hepatocellular carcinoma (HCC) and drives increased IL-23 production by inducing STAT3 phosphorylation to confine hepatocyte nuclear factor 4 alpha (HNF4α) [39].

In Adeno-associated virus-infected (AAV-infected) patients, the IL-23/IL23R blockade inhibits a series of responses including the stimulation of soluble IL-23 (sIL-23), which hinders STAT3 phosphorylation in the cytoplasm [40]. The level of IFN-γ is also downregulated via inhibition by the IL-23/IL-17 axis of Th17 cells [40]. Interestingly, endogenous IL-23 is a source of disease with or without IL-17 action, while exogenous IL-23 requires IL-22 to protect patients from pneumonia [34]. In contrast, in chronic hepatitis B patients, chronic HBV induces IL-1/6/17 and transforming growth factor-β1 (TGF-β1) but not IL-23 [33].

Overall, IL-23 has great potential for use in inflammatory autoimmune disease treatment [41].

### 2.3. IL-27

As a factor that suppresses inflammation, IL-27 can modulate IL-10 secretion by Tr1 via STAT1, but IL-27 can also work as a pro-inflammatory cytokine to break down CD4+Tregs or by activating Th1 differentiation [2]. Like IL-12, IL-27 is essential to induce naïve CD4+ cells and promotes the transformation of NK cells into Th1 cells to produce IFN-γ, together with activating T-bet and IL-12R [24]. Recently published results indicate that TLRs stimulate the expression of *MyD88*, leading to an optimal production of IL-12, IL-23, and IL-27 [42]. Without *MyD88*, Th1 responses are abolished and Th2 responses are enhanced [43]. For instance, during HIV-68 infection, the TLR4 agonist stimulates p28 mRNA expression [44], thus increasing Th1 cells. However, many studies have focused on HIV-infected patients, and the findings show that the production of IL-27 and the expression of p28 mRNA are both inhibited in LPS-stimulated human monocyte-derived MΦs (MDMs). MDMs, with the help of trans-activating protein (Tat), induce a decrease in p38 mitogen-activated protein kinase (p38 MAPKs) [45] (Table 1). IL-27 is not directly responsible for HIV-1 inhibition in CD4 cells but stimulates a type I IFN-independent mechanism after viral entry [46]. Recently, highly active antiretroviral therapy (HAART) has been recognized as a potential approach toward treating HIV infection. After 6 or 12 months of this treatment, data show that the concentrations of plasma IL-27 and gp130-positive cells are positively correlated with numbers of peripheral blood CD3+CD4+ cells but negatively associated with plasma HIV viral load [46]. Additionally, IL-27 can promote hematopoietic stem cell (HSC) diffusion, cytotoxic lymphocyte (CTL) production, and TH1 cell differentiation [47]. Hence, IL-27 and its receptors have been shown to be effective in HIV treatment, and the Th1/Th2 ratio is a latent influencing factor since Th1 induction is relevant to levels of CD4+ cells [48]. Similarly to HIV infection, HCV infection also induces IL-27 production, which blocks HCV replication and initiates the expression of antiviral genes [49]. However, when HIV-1 and HCV coinfect a cell, the function of IL-27 appears to be weakened [50].

During influenza A virus (IAV) infection, cyclooxygenase-2 (COX-2) and the protein kinase A-cAMP response element binding protein (PKA-CREB) signaling pathway (both calcium-PKA-CREB and COX-2/PGE2-PKA-CREB signaling pathways) [51] can activate IL-27 expression [52]. This eventually inhibits IAV replication in influenza patients [51]. COX-2 promotes prostaglandin E2 (PGE2) accumulation, which leads to the activation of the IL-27/EBi3 promoter [51]. Simultaneously, with stimulation of gp130, a subunit of IL-6R, STAT3 can activate the expression of IL-6 [53]. In addition, the soluble IL-6R (sIL-6R) and IL-27 subunit p28 complex is closely associated with IFN-λ1 expression by regulating the p38 MAPK signaling pathway. When this pathway is initiated, the amount of c-Fos and phosphorylated activating transcription factor 1 (ATF1) markedly increases. These two factors tend to form a heterodimer and then bind to the IFN-λ1 promoter region, which includes the cis element AP-1, to activate IFN-λ1 expression [54].

In HBV infections, IL-27 favors the phosphorylation of STAT1/3; the expression of T-bet; the inhibition of GATA binding protein 3 (GATA3), which participates in Th2 differentiation, and the production of IL-12Rβ2; and triggers naïve T cells to differentiate into Th1 cells [9,55]. In contrast to its function in repressing HBV gene expression and clearing viruses, IL-27 can also avoid excessive T-bet activation, leading to a decrease in IL-12Rβ2 [9,56]. IL-27 accounts for a delay in virus control by promoting virus-specific Tr1-like CD4+T cells, which not only limits all inflammation but also antiviral T cell activity [57] (Table 1). 

According to experiments on chronic HBV infection, IL-27 markedly increases and directly affects inflammatory responses in a positive pathway by synergizing with other antiviral cytokines [58]. Available data indicate that the amount of IL-27 is closely related to the level of liver damage. This is consistent with the fact that IL-27 is a major pro-inflammatory factor in HBV infection [59]. Although high levels of IL-27 have little effect on Th1/2 cells, they exert a negative feedback on T cells [59]. The presence of IL-27 can directly promote type III IFN (IFN-λ1) production, which activates IFN-λ1 receptors through ERK1/2 signaling and enhances NF-κb nuclear transduction (Table 1). Because IL-27 has similar functions to IFN-α, IL-27 can also regulate interferon-sensitive responsive element (ISRE), which results in expression of IFN-stimulated genes and ultimately inhibits the production of HBV protein and the replication of capsid-associated genes [52]. As described above, IL-27p28 is produced faster than EBi3, and the production of IL-27 requires the co-expression of p28 and EBi3. IL-35 competes with IL-27 due to competition for EBi3, and IL-35 can significantly increase after 24 h of IAV infection [51]. This is not what occurs in EBV infection, because CD40 cells induce EBi3 expression in B cells when NF-κb is present, and then EBi3 functions as both a pro-Th1 and a pro-Th2 factor [60].

Overall, it is obvious that IL-27 has a strong antiviral capacity and is involved in various inflammatory networks in infected organisms.

### 2.4. IL-35

In contrast to the cytokines discussed above, increases in IL-35 are due to the presence of regulatory T cells (Tregs) and regulatory B cells (Bregs) [61]. Treg cells can promote the production of new Treg cells by inducing the transformation of Treg (iTreg) from CD4+Foxp3-T cells [62]. IL-35-induced Treg cells (iTr35) produce more IL-35 [62], and type 1 Treg cells (Tr1) use IL-35 to suppress the immune response [63]. Unlike IL-12 and IL-23, IL-35 suppresses inflammatory responses by regulating various cytokines, thereby controlling STAT signaling [64]. As a result, the binding of IL-35 with IL-35R can activate STAT1 and STAT4 in T cells and trigger STAT1 and STAT3 in B cells [65]. IL-35 inhibits monocyte-derived dendritic cell (MoDC) maturation by triggering the STAT 1/3 pathways and simultaneously blocking p38 MAPK and NF-κb signaling pathways, consequently down-regulating pro-inflammatory functions [66].

Counteracting the effect of IL-12, which plays an important role in providing the positive feedback that links T cells and IFN-γ, IL-35 suppresses T cell proliferation. IL-35 causes great harm to CD8+T cells, leading to the breakdown of immunity or immune dysfunction [64]. Additionally, there is evidence that suggests that a deficiency in IL-10 enhances IL-35 expression, which could suppress Th2 cell differentiation and cytokine elaboration [67]. Another interesting phenomenon is that CD4 T cells induce the expression of IL-35, causing a decrease of IFN-γ, while CD4 T cell-induced IL-12 dramatically increases IFN-γ [68].

Due to IFN-γ-induced negative regulation, IL-35 is not conducive to Th17 and Th1 cell proliferation and counters the activity of IFN-γ and TNF-α through STAT1 [69] (Figure 1) during HBV infection [65,70] (Table 1). Furthermore, IL-35 promotes HBV replication and transcription by targeting HNF4α, resulting in an increase of HBV 3.5 kb mRNA levels, HBV core protein levels, and hepatitis Be antigen (HBeAg) secretion levels [68]. To summarize, IL-35 functions in chronic HBV infection improves immune tolerance through the inhibition of pro-inflammatory cytokine expression [70]

Similarly, IAV also enhances IL-35 in peripheral blood mononuclear cells, throat swabs [71], human primary lung cells, and PBMCs that induce IFN-γ expression [70]. Unlike HBV infection, NF-κb is the key component that mediates IL-35 transcription during IAV infection, and IL-35 may suppress IAV protein synthesis by inducing type I and type III IFN [71]. Furthermore, STAT1 and STAT4 signaling pathways must be activated for effective function of IL-35.

Consideration of IL-35 is absolutely necessary in the field of liver damage treatment, because it can interfere with the process of HBV infection as well as chronic HCV infection. In chronic hepatitis C virus infection, IL-35 production increases Foxp3 mRNA expression and anti-inflammatory factors so that the control of immune responses leads to persistent damage to the liver [72]. Recent studies show that CD4+CD25+Foxp3+ Tregs cells that can suppress immune systems are induced by AdIL-35 [73]. However, STAT1 and STAT4 signaling pathways must be activated to allow IL-35 to function smoothly.

There are still only a few studies on IL-35, so the functions of IL-35 in combatting viral infection are not yet well understood.

## 3. Conclusions

The existing literature reveals the importance of the IL-12 family of cytokines, including IL-12, IL-23, IL-27, IL-35, and IL-39. These cytokines are similar in structure but not in their functions. Although these cytokines play a role in resistance to both viruses and bacteria, this review focuses on their activity against viral infection. IL-12 and IL-23 mainly play roles as pro-inflammatory cytokines, while IL-27 can be both pro- and anti-inflammatory in its effects, and IL-35 suppresses immunity by Tregs (Figure 2). IL-39, a newly-identified recombinant cytokine, has yet to have its roles in immune responses and viral infection elucidated. T cell proliferation and the innate immune response are intensively involved in the regulation of the IL-12 family. IFN-α, IFN-γ, and IFN-λ1 promote the activity of T lymphocytes, which are related to Th cell differentiation, so that the immune system is activated to defend against viral infection. RANKL works with NF-κb and TNF-α, having remarkable anti-viral effects. Investment and efforts to develop vaccines based on cytokine action are warranted, as attenuated vaccines have prevented humans from several viral infections, and other forms of vaccines may realize the health maintenance of human beings. Another promising strategy is to combine different subunits of IL-12 family cytokines to create new, recombinant cytokines. In conclusion, this review elucidates the role of IL-12 family members during the process of combatting different viruses. Specific IL-12 family agonists and antagonists deserve further attention, especially from the perspective of disease pathogenesis. Whether in innate or acquired immune responses, optimal regulation of the in vivo interaction networks of cytokines is an important new stage of research. 

## Figures and Tables

**Figure 1 viruses-11-00772-f001:**
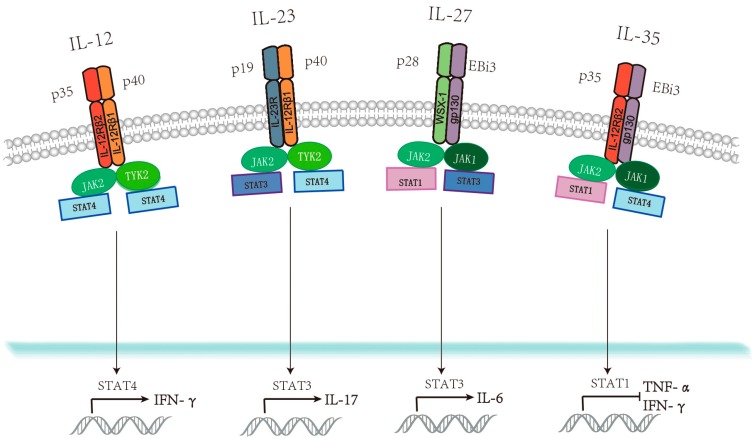
Interleukin 12 (IL-12) family members, corresponding receptors, and regulation of downstream signaling pathways.

**Figure 2 viruses-11-00772-f002:**
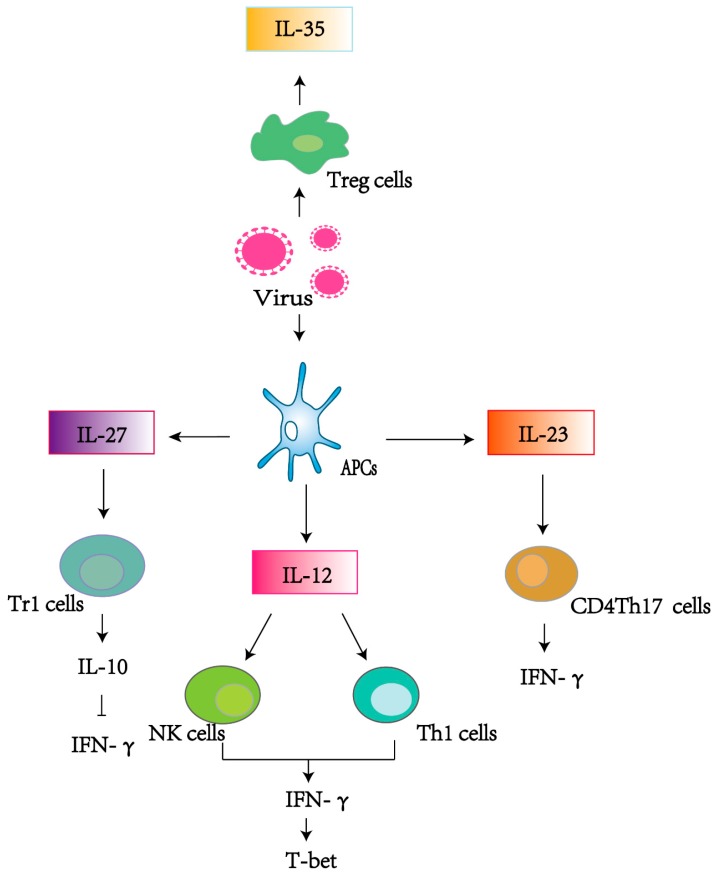
IL-12 family members mediate the immune responses when virus enters into cells. Viral infection contributes to expression of IL-12, IL-23, IL-27, and IL-35 by activating antigen-presenting cells (APCs) or regulatory T (Tregs) cells. IL-12, IL-23, and IL-35 are active in the promotion of inflammatory responses, and IL-27 can function as a kind of anti-inflammatory cytokine. Overall, the IL-12 family of cytokines is indispensable for virus clearance by regulating IFN-γ and other cytokines.

**Table 1 viruses-11-00772-t001:** Immune regulation by IL-12 family members during hepatitis B virus (HBV) and human immunodeficiency virus (HIV) infection.

	*HIV*	*HBV*
***IL-12***	CD40L+TNF-α 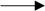 IL-12 Th1 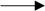 IFN-γ 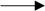 T-bet 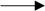 IL-12R IFN-α2b, RANKL suppress HIV	IL-12 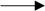 CD4/8 T cells 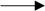 IFN-γ TNF-α, T-bet suppress HIV
***IL-23***	Th17 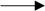 IL-17/22 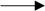 IL-23 NK 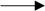 IL-23R IFN-α, SOCS1 suppress HIV	IL-23 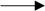 IL-17 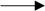 γδT cells IL-23 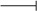 IFN-γ, TNF-α, SOCS1 Th17, PD-1 suppress HIV
***IL-27***	MDMs + Tat 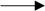 p38 MAPKs IFN-λ1 suppresses HIV	IL-27 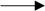 Tr1-like CD4 T cells IL-27 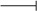 GATA3, T-bet IFN-λ1, NF-κb suppress HIV
***IL-35***	Unknown	IL-35 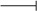 Th1/17 IL-35 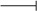 IFN-γ, TNF-α Promoting HBV

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
