# Peer review of "Immunoregulatory Functions of the IL-12 Family of Cytokines in Antiviral Systems"

_viruses, 2019, doi:10.3390/v11090772_

Round 1

Reviewer 1 Report

This review from Guo et al. is interesting and relevant for readers that are novices or experts in immunology. However, the review lacks a sense of direction at times and may not be logically organized. It often reads like a long list of facts that appear to be described in random order. Given the journal choice (Viruses), more text about viruses and the antiviral activities of the IL-12 family should appear in the Introduction.

Line 65: Remove “their” before “production.”

Line 68: It’s unclear what the term “induces” means in this sentence. Activates?

Line 99: Insert “an” before “apparent.”

Line 104: Again, it is unclear what the term “inducible” really means here.

Line 134: “Another review.” Instead of just citing the review when making this statement, the original primary publication that made the discovery should be cited.
